# A Unified View of Finding and Transforming Winning Lottery Tickets

## Abstract

While over-parameterized deep neural networks obtain prominent results on various machine learning tasks, their superfluous parameters usually make model training and inference notoriously inefficient. Lottery Ticket Hypothesis (LTH) addresses this issue from a novel perspective: it articulates that there always exist sparse and admirable subnetworks in a randomly initialized dense network, which can be realized by an iterative pruning strategy. Dual Lottery Ticket Hypothesis (DLTH) further investigates sparse network training from a complementary view. Concretely, it introduces a gradually increased regularization term to transform a dense network to an ultra-light subnetwork without sacrificing learning capacity. After revisiting the success of LTH and DLTH, we unify these two research lines by coupling the stability of iterative pruning and the excellent performance of increased regularization, resulting in two new algorithms (UniLTH and UniDLTH) for finding and transforming winning tickets, respectively. Unlike either LTH without regularization or DLTH which applies regularization across the training, our methods first train the network without any regularization force until the model reaches a certain point (i.e., the validation loss does not decrease for several epochs), and then employ increased regularization for information extrusion and iteratively perform magnitude pruning till the end. We theoretically prove that the early stopping mechanism acts analogously as regularization and can help the optimization trajectory stop at a particularly better point in space than regularization. This does not only prevent the parameters from being excessively skewed to the training distribution (over-fitting), but also better stimulate the network potential to obtain more powerful subnetworks. Extensive experiments show the superiority of our methods in terms of accuracy and sparsity.

## 1 Introduction

Exactly as saying goes: *you can't have your cake and eat it* – though over-parameterized deep neural networks achieve encouraging performance over widespread machine learning tasks Zagoruyko & Komodakis (2016); Arora et al. (2019); Devlin et al. (2018); Brown et al. (2020), they usually suffer notoriously high computational costs and necessitate unaffordable storage resources Cheng et al. (2017); Deng et al. (2020); Wang et al. (2019a). To alleviate this issue, a stream of pruning approaches Han et al. (2015); Liu et al. (2017); He et al. (2017); Gale et al. (2019); Ding et al. (2019) tries to uncover a sparse subnetwork that can retain the learning capacity of the original dense network as much as possible. While these algorithms seek to reach a preferable trade-off between performance and sparsity, they fall short of satisfying the joint optimization of both.

Recently, Lottery Ticket Hypothesis (LTH) has provided a novel perspective to investigate sparse network training Frankle & Carbin (2018). It articulates that there consistently exist sparse and high-performance subnetworks in a randomly initialized dense network, like winning tickets in a lottery pool. To identify such admirable sparse subnetworks (i.e., winning tickets), LTH trains an over-parameterized neural network from scratch and prunes its smallest-magnitude weights iteratively, which is so called *iterative pruning*. This repeated pruning method, as opposed to one-shot pruning, allows us to learn faster and achieve higher test accuracy at smaller network size. LTH innovatively exposes the internal relationships between a randomly initialized network and its corresponding subnetworks, inspiring a series of follow-ups to explore various iterative pruning and rewind criteria for training light-weight networks Morcos et al. (2019); Maene et al. (2021); Chen et al. (2021); Frankle et al. (2019; 2020); Ding et al. (2021); Ma et al. (2021); Chen et al. (2022).

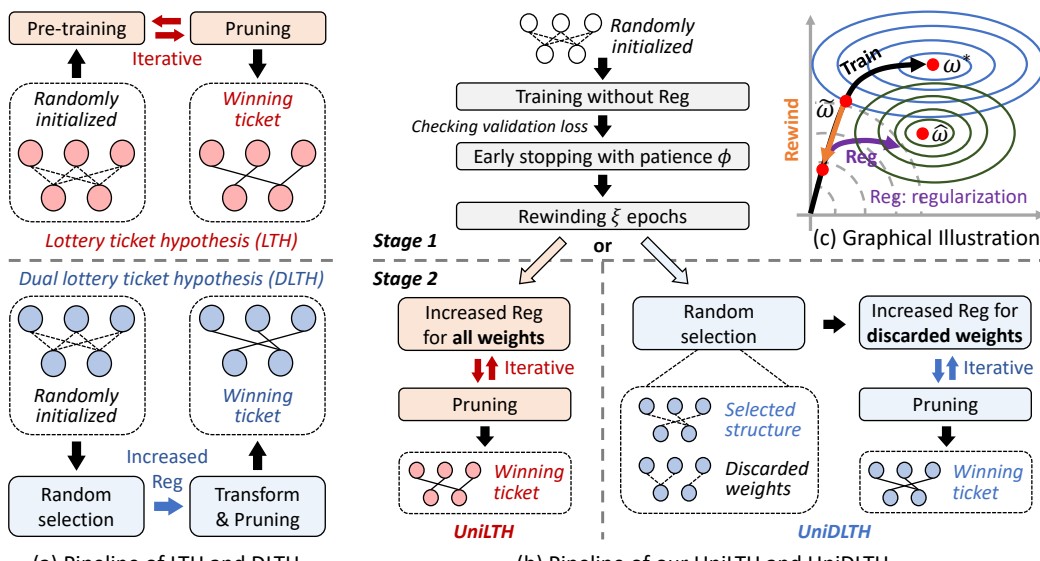

Figure 1: Illustration of LTH/DLTH and our UniLTH/UniDLTH. In (c), the blue/green solid contour lines denote the contours of the training/validation negative log-likelihood. Our goal is to drew the weights closer to $\hat{w}$. The black line indicates the training trajectory taken by SGD. Our algorithm rewinds the training procedure (the yellow line) and add increased regularization (the purple line) to move towards the validation set distribution when training reaches an early stopping threshold.

Though promising, LTH concentrates solely on identifying one sparse subnetwork by iterative pruning, which is not universal to both practical usages and investigating the relationship between dense networks and its subnetworks Bai et al. (2022). Hence, Bai et al. (2022) go from a complementary direction to propose Dual Lottery Hypothesis (DLTH) which studies a randomly selected subnetwork rather than a particular one. As a dual problem of LTH, it hypothesizes that a randomly selected subnetwork in a randomly initialized dense network can be turned into an appropriate condition with excellent performance, analogy to *transforming* a random lottery ticket to a winning ticket. To validate this, DLTH trains a dense network and conducts one-shot pruning with a simple yet effective strategy – it identifies the sparse subnetwork by utilizing a *gradually increased regularization term* throughout the training phase, which extrudes information from *unimportant* weights (which will be pruned afterward) to target a sparse neural structure. Although this hypothesis does not provide any theoretical proof on how much information extrusion we can achieve, it does provide a novel view on harnessing regularization terms to link the dense network with hidden winning tickets.

As the key element to DLTH's success, the regularization term realizes information extrusion from the unimportant weights which will be masked (i.e., discarded), but may also become its undoing. In a training process, the equilibrium of all network weights is usually determined by two forces: loss gradient force and regularization gradient force. The latter one is generally maintained in a small regime, as the excessive weight penalty will cause the network to collapse into a suboptimal local minimum, corresponding to the ill-conditioned small weights LeCun et al. (2015). Using a regularization term at the early training phase as DLTH does, may cripple the model performance since it complicates the network optimization and misleads the finding of a reliable equilibrium. Meanwhile, regularization-based pruning approaches (e.g., DLTH) typically perform one-shot pruning, which exacerbates the instability of sparse network training. Considering the efficacy of iterative pruning in LTH, transforming random tickets into winning tickets iteratively is appealing as well.

In this paper, we aim at presenting a resilient and unified paradigm for searching winning tickets in a dense network (LTH) or transforming random tickets to winning tickets (DLTH), leading to two new pruning algorithms termed **UniLTH** and **UniDLTH**. As illustrated in Fig. 1(b), both UniLTH and UniDLTH decouple the pruning task into two separate stages. At the first stage, the two algorithms share an identical procedure – they do not set up any obstacle force (regularization) when training a randomly initialized network. Once the validation loss does not decrease for several training cycles, we cut off the training and rewind the network parameters to several epochs earlier. We demonstrate that utilizing such an *early stopping* strategy can defeat the instability caused by regularization, thus achieving similar or even better performance without compromising the learning potential.

At the second stage, we integrate the iterative pruning with the increased regularization for searching or transforming winning tickets. To be more specific, we alternately train the network with increased regularization and perform pruning to tilt the network distribution towards the validation distribution until the network reaches the corresponding sparsity. The major difference between UniLTH and UniDLTH is the weights to which we apply regularization (see Fig. 1(b)). UniLTH differentiates the magnitudes of the weights by applying progressively increasing regularization on all the weights, while UniDLTH applies $L_2$ regularization only on the unimportant weights for information extrusion. The contributions of this paper can be summarized as follows:

- We introduce a unified view of searching and transforming winning lottery tickets. We find that the removal of regularization force of the early training phase is more helpful for preserving network expressivity. This simple yet efficient winning ticket search/transform paradigm can seamlessly translate to arbitrary networks without being subject to specific network structure.

- We provide a theoretical proof of the early stopping strategy's ability to substitute $L_2$ regularization, as well as an intuitive explanation of its benefits. This new training paradigm show great promise in retrieving winning tickets from a large dense network. We also verify that networks can perform better under lower regularization pressures with a novel nonlinear regularization scheme.

- We conduct extensive experiments to evaluate our algorithms in terms of sparsity and performance. In particular, UniLTH outperforms LTH by 0.19%~1.35% and UniDLTH surpasses DLTH 0.29%~0.56% on accuracy over four representative backbones on the CIFAR-10 dataset. Remarkably, we can even obtain 90% sparse winning tickets without performance dropping, especially on the large-scale dataset ImageNet, which verifies the superiority of our algorithms.

## 2    RELATED WORK

**Winning Lottery Tickets.** LTH elaborately draw an analogy between uncovering admirable subnetworks in a dense network and finding winning tickets in a lottery pool. It articulates that *a randomly initialized dense network contains a high-performance subnetwork which can be trained in isolation for at most the same number of iterations as the original network Frankle & Carbin (2018)*. In light of LTH, some follow-ups have explored the prospect of training sparse subnetworks in place of the entire models without sacrificing performance Malach et al. (2020). LTH is also adopted to discover the presence of "supermasks", which can transform an untrained, randomly initialized network to a far higher-performance model Zhou et al. (2019). Furthermore, a stream of research has been dedicated to the findings of early-bird (EB) tickets You et al. (2019) (the tickets which emerge at the very early training phase) to reduce the computational overheads, e.g., in graph neural networks You et al. (2021); Chen et al. (2021), in natural language processing Chen et al. (2020). To generalize the lottery tickets across tasks, Zhou et al. (2019); Morcos et al. (2019) verify the performance of the winning ticket initializations generated by sufficiently large datasets, and discover these subnetworks contain inductive biases generic to neural networks more broadly which improve training across many tasks. In addition to the above work, DLTH considers a more general and challenging case to find the relationship between a dense network and its sparse counterparts Bai et al. (2022) – *it argues that a randomly selected subnetwork from a dense network can be transformed into a trainable condition and achieve admirable performance compared with the winning lottery pool Bai et al. (2022), which suggests a more adjustable way of investigating sparse neural networks.*

**Regularization-based Pruning.** Regularization has long been exploited for pruning deep neural networks by enforcing a part of parameters in the original network to zeros. The most popular approaches are using $L_0$-norm or $L_1$-norm Louizos et al. (2017); Liu et al. (2017); Ye et al. (2018). For instance, He et al. (2017) adopt LASSO to achieve channel pruning for accelerating very deep neural networks. Following this trend, the Group LASSO algorithm is further introduced to obtain a regular sparse subnetwork Lebedev & Lempitsky (2016); Wen et al. (2016). Ding et al. (2018) proposes to employ various penalty factors for different weights. Among these works, the "regularization force" is maintained in a small regime to avoid crippling the model performance. To take advantage of the model sparseness brought by the large penalty strength, Wang et al. (2019b; 2020) present the first attempt to utilize gradually increased regularization terms to achieve high sparsity and preserve the admirable performance of the original model. However, the magnitude of the regularization term is extremely important and needs to be carefully controlled, since an excessive weight penalty will may cause the model weights ill-conditioned. In this paper, we focus on a simple and efficient usage of the regularization term to discover more reliable subnetworks, i.e., winning tickets, while ensuring that the regularization term does not have a catastrophic influence on the training process.

## 3 METHODOLOGY

### 3.1 A UNIFIED APPROACH FOR SOLVING LTH AND DLTH

Here, we delineate our algorithm which unifies the line of LTH and DLTH to obtain winning lottery tickets. As depicted in Algo. 1, it can be flexibly adopted to the settings of LTH and DLTH with a few modifications. We denote in blue the procedures for uncovering the winning tickets (marked as UniLTH), and in red the procedures for transforming a random selected subnetwork to the winning tickets (i.e., UniDLTH). The other lines (in black) are shared by both.

**UniDLTH.** We first describe UniDLTH which aims to *transform randomly-selected tickets to the winning tickets*. Unlike DLTH, we do not introduce any regularization force at the early training phase to ensure that the model accurately learns the training data distribution. Instead, we perform training, early stopping, and rewind at the first stage (Line 1-6) – when the validation loss does not drop in $\phi$ epochs, we stop training and rewind the parameters to $\xi$ epochs earlier. Then we produce the random tickets by randomly selecting a set of important parameters (the rest are unimportant) in Line 7. After rewinding all weights, we integrate the iterative pruning strategy with the gradually increased regularization (Line 8-16) to extrude information from unimportant weights $\Theta_{un}$ to target the sparse structure, i.e., the winning tickets $\Theta_{im}$.

**UniLTH.** For UniLTH, our target is to *find the winning tickets in a dense, random initialized neural network*. The whole pipeline is shown in Algo. 1 and almost similar to UniDLTH, thus we mainly highlight how UniLTH differs from UniDLTH. Firstly, UniLTH does not need to specify a random set of weights to realize information extrusion as UniDLTH does at the second stage (in Line 7). Secondly, UniLTH applies regularization to the universe of the network while UniDLTH merely decays the unimportant weights (see Line 9). Lastly, UniDLTH prunes the non-important weights at each iteration, whereas UniLTH has no such restriction (see Line 13).

As described, our pruning algorithm is iterative-based. Suppose a dense network has been pruned for $i$ iterations and each round cut off $p\%$ of the weights that survive in the previous iteration, we target a sparse structure to be $r\%$ of the original network size. In this case, $p$ can be expressed as $p\% = 1 - (r\%)^{1/\psi}$ and decreased over iterations. Notably, such pruning rate becomes very low near the end of the pruning, which is analogy to the fine-tuning technique. The rest of this section are organized as follows. In Sec. 3.2, we will theoretically prove the equivalence between early stopping and regularization, and explain why we should employ early stopping and rewinding rather than regularization at the early stage. In Sec. 3.3, we will elaborate on the second stage, i.e., iterative pruning with three variants of nonlinear increased regularization.

---

**Algorithm 1** UniLTH and UniDLTH Algorithms (aligned with Fig. 1)

**Require:** Network $f(X, \Theta)$ with data $X$ and parameters $\Theta$; sparsity level $S_f$; mask matrix $m_\Theta$ with initialization $m_\Theta^0 = 1$; pruning rate $p\%$; times $\psi$; step size $\eta$; patience $\phi$ for early stopping.
1: **while** $1 - \frac{||m_\Theta||_0}{|m_\Theta^0|} < S_f$ **do**
2:     Forward $f(X, \Theta)$ to compute loss $\mathcal{L}_s = \mathcal{L}(X, \Theta)$
3:     Update $\Theta^{(j+1)} \leftarrow \Theta^{(j)} - \eta \nabla_{\Theta^{(j)}} \mathcal{L}_s$
4:     **if** the validation loss is not descending for $\phi$ epoch **then**
5:         Rewind $\Theta$ to several training epochs ago (Note weights as $\Theta^{(E)}$)
6:     Load weight $\Theta^{(E)}$
7:     Select unimportant weights $\Theta_{\text{un}}$ and important weights $\Theta_{\text{im}}$
8:     **for** iteration $i = 1, 2 \ldots M$ **do**
9:         Forward $f(X; m_\Theta; \Theta)$ to compute the loss $\tilde{\mathcal{L}}_s = \mathcal{L}_s(X, m_\Theta \odot \Theta) + \alpha ||\Theta||_2^2$ (or $\tilde{\mathcal{L}}_s = \mathcal{L}_s(X, m_\Theta \odot \Theta) + \alpha ||\Theta_{un}||_2^2$), where $\alpha$ is a hyper-parameter to control sparsity
10:         Update $\Theta^{(i+1)} \leftarrow \Theta^{(i)} - \eta \nabla_{\Theta^{(i)}} \tilde{\mathcal{L}}_s$
11:         Adopt nonlinear increasing regularization (i.e., LogLaw, TanLaw and ExpLaw. )
12:         **if** $i == M/\psi \cdot \gamma (\gamma = 1, 2 \ldots \psi)$ **then**
13:             Prune $p\%$ parameters with the lowest magnitude values in $\Theta$ (or $\Theta_{\text{un}}$)
14:             Update $m_\Theta$ by zeroing the elements in $m_\Theta$ which are corresponding to the $p\%$ pruned parameters
15:         **if** $1 - \frac{||m_\Theta||_0}{|m_\Theta^0|} < S_f$ (or $\Theta_{un} = 0$) **then**
16:             Stopping training and obtain $\Theta^{(M)}$
17: **return** $m_\Theta \odot \Theta^{(M)}$ or $\Theta_{im}$

---

## 3.2 EARLY STOPPING VERSUS REGULARIZATION

As the key operation of DLTH, regularization borrows learning capacity and perform information extrusion from the pruned weights from beginning to end. However, both *early* and *excessive* regularization considerably impedes the model expressivity: 1) At the early stage, the model should focus more on fitting the data distribution without restricting its capacity. 2) Excessive regularization force makes the training process hard to control, e.g., a slight excess may cause irreversible reactions or ill-conditioned weights. To this end, we figure out that the early stopping mechanism is a simple yet effective alternative to regularization at the early training phase with a *theoretical proof*. Compared to regularization, early stopping not only has no negative effect on the early-stage behavior of fitting to the data distribution, but is also more controllable due to its non-parametric nature. In next part, we will mathematically analyze why regularization is equivalent to early stopping.

$L_2$ **Regularization.** We first revisit the theory of regularization. Let $J(w)$ denote an *unregularized* objective function, $w^* = \mathrm{argmin}_w J(w)$ is the weight vector when $J(w)$ achieves the minimum training error. Assuming the existence of second-order partial derivatives, we perform a quadratic approximation to the unregularized objective function in a small neighborhood of $w^*$ as

$$\hat{J}(w) = J(w^*) + \frac{1}{2}(w - w^*)^T H (w - w^*), \tag{1}$$

where $w$ in the neighborhood of $w^*$ and $H$ is the Hessian matrix LeCun et al. (1990) at point $w^*$. Given $w$ is a local optimal point, the first-order term (Jacobian matrix) in Eq. 1 has been eliminated and $H$ is positive semi-definite. When $\hat{J}(w)$ get minimum, we have $\nabla_w \hat{J}(w) = H(w - w^*) = 0$.

$L_2$ regularization has long been the most popular form of regularization, also known as weight decay Cortes et al. (2012), which ensures the weight constrained within a small range by adding a penalty term $\Omega(w) = \frac{1}{2}\alpha||w||_2^2$ to $J(w)$. Under this circumstance, the optimal point of $w$ will be perturbed by the new force, causing the network to reach a new balance point $\acute{w}$ as

$$\alpha\acute{w} + H(\acute{w} - w^*) = 0 \quad \Rightarrow \quad \acute{w} = (H + \alpha I)^{-1} H w^*. \tag{2}$$

With the increase of $\alpha$, we decompose $H$ ($H$ is real symmetric) into the diagonal matrix $\Lambda$ and the standard orthonormal basis $Q$ of the eigenvectors, $H = Q\Lambda Q^T$:

$$\acute{w} = (Q\Lambda Q^T + \alpha I)^{-1} Q\Lambda Q^T w^* = Q(\Lambda + \alpha I)^{-1}\Lambda Q^T w^*. \tag{3}$$

As seen above, the weight decay is essentially scaling $w^*$ along the axis defined by the eigenvectors of $H$. This scaling effect has less effect on the direction with larger eigenvalues and more on the direction with smaller eigenvalues, indicating that the weights of various curvatures in the network behave differently when performing such regularization.

**Early Stopping.** A fundamental principle is recognized in deep neural networks – *when the validation set loss does not decrease in several training cycles, we can intuitively find that the model has undergone slight over-fitting.* We usually stop training as soon as the error on the validation set is higher than it was the last time it was checked, which is so called early stopping. Compared with $L_2$ regularization, early stopping is more inconspicuous since it barely affects the training process, the objective function, or any set of allowable parameter values.

Suppose $J(w)$ reaches its minimum (see Eq. 1), we obtain $\nabla_w \hat{J}(w) = H(w - w^*) = 0$. We study the process of parameter updates during training, assuming we optimize $\tau$ steps with the learning rate $\epsilon$, by analyzing the gradient descent on $\hat{J}$ to approximately study the gradient descent on $J$. The parameters of the $\tau$-th step can be derived from the $(\tau - 1)$-th step as follows:

$$w^{(\tau)} = w^{(\tau-1)} - \epsilon H\left(w^{(\tau-1)} - w^*\right) \quad \Rightarrow \quad w^{(\tau)} - w^* = (I - \epsilon H)\left(w^{(\tau-1)} - w^*\right) \tag{4}$$

Taking $H = Q\Lambda Q^T$ into the above equation, we can easily obtain:

$$\begin{cases} Q^T\left(w^{(\tau)} - w^*\right) = (I - \epsilon\Lambda) Q^T\left(w^{(\tau-1)} - w^*\right) \\ Q^T\left(w^{(\tau-1)} - w^*\right) = (I - \epsilon\Lambda) Q^T\left(w^{(\tau-2)} - w^*\right) \\ \qquad\qquad\qquad \vdots \\ Q^T\left(w^{(1)} - w^*\right) = (I - \epsilon\Lambda) Q^T\left(w^{(0)} - w^*\right) \end{cases} \Rightarrow \begin{aligned} Q^T w^{(\tau)} &= (I - \epsilon\Lambda)^\tau Q^T w^{(0)} + \\ &\quad [I - (I - \epsilon\Lambda)^\tau] Q^T w^* \end{aligned}$$

$$\tag{5}$$

For simplicity, we first prove the equivalence under $w^{(0)} = 0$. As a result, Eq. 5 can be simplified into $Q^T w^{(\tau)} = \left[ I - (I - \epsilon\Lambda)^\tau \right] Q^T w^*$. For $L_2$ regularization (see Eq. 3), $\Lambda$ can be written as $\Lambda = \mathrm{diag}(\lambda_1, \ldots, \lambda_n)$, then we can get an expression similar to the early stopping:

$$Q^T \acute{w} = (\Lambda + \alpha I)^{-1} \Lambda Q^T w^* = \mathrm{diag}(\frac{\lambda_1}{\lambda_1 + \alpha}, \cdots, \frac{\lambda_n}{\lambda_n + \alpha}) Q^T w^* = \left[ I - (\Lambda + \alpha I)^{-1}\alpha \right] Q^T w^* \tag{6}$$

Now we are able to link the formula of $L_2$ regularization (Eq. 6) with the early stopping (Eq. 5):

$$\underbrace{Q^T \acute{w} = \left[ I - (\Lambda + \alpha I)^{-1}\alpha \right] Q^T w^*}_{L_2 \text{ Regularization}} \quad \Leftrightarrow \quad \underbrace{Q^T w^{(\tau)} = \left[ I - (I - \epsilon\Lambda)^\tau \right] Q^T w^*}_{\text{early stopping}} \tag{7}$$

As shown in Eq. 7, the early stopping is equivalent to $L_2$ regularization if $(\Lambda + \alpha I)^{-1}\alpha = (I - \epsilon\Lambda)^\tau$ satisfies. In this case, we further take the logarithm form on both sides and derive:

$$\tau \log(I - \epsilon\Lambda) = -\log(I + \Lambda/\alpha) \tag{8}$$

When $\tau\epsilon\Lambda \approx \frac{\Lambda}{\alpha}$ (i.e., $\alpha \approx \frac{1}{\epsilon\tau}$), $L_2$ regularization is equivalent to the early stopping mechanism. More generally, when $w^{(0)} \neq 0$, we can draw similar conclusions (The above two proofs will be shown in detail in Appendix A and B). $\epsilon\tau$ is the product of the learning rate and the number of steps, which can be intuitively regarded as the capacity of the network. For some changing learning rates, $\epsilon\tau$ can be viewed as the distance that the optimization curve moves in the high-dimensional space.

**Why Early Stopping is Preferable?** After demonstrating their equivalence, we articulate the major advantages of our early stopping strategy against regularization. In the case of early termination, this actually means in the direction of the large curvature parameters than smaller curvature direction earlier to learn, at the same time can terminate automatically determine the result of the regularization (only need to change the observation test loss decline times), and the weight decay require different parameter values of weights training experiment. We believe that excessive reliance on regularization at the early stage of training is not conducive to the network moving to the training distribution. At the early stage of training, we only use the early stopping strategy, which can not only learn the data distribution better, but also will not disrupt the dynamic learning process of the network.

### 3.3 ITERATIVE PRUNING WITH INCREASED REGULARIZATION

We perform our pruning procedure *iteratively* after rewinding the weights to several epochs earlier, i.e., we repeatedly train, prune, and reset the network across $M$ rounds (i.e., iterations). Each round eliminates $p\%$ of the weights that survive the previous round, indicating that the remaining weights in the final round has been constantly verified for $M$ times, which ensures the reliability of the retained weights. In comparison to one-shot pruning, this iterative strategy empowers us to learn faster and achieve higher test accuracy with a small-scale network (see our experiments in Sec. 4.1).

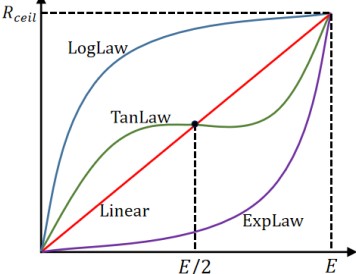

Figure 2: Different nonlinear increased regularization methods. For simplicity, we draw these nonlinear growth curves in continuous forms.

During the pruning process, we perform increased regularization to tilt the network distribution towards the validation distribution. Existing regularization-based pruning approaches (e.g., DLTH) mostly utilize linear increased regularization, where the coefficient $\alpha$ grows by a constant for each round. As a result, the regularization will increase continuously at equal intervals to mine out the diversity of parameters Bai et al. (2022) and enable us to obtain a more expressive subnetwork (see the proof in Appendix C).

However, linear increased regularization may not be conducive to the search of optimal parameters, since a large magnitude of regularization may destroy the network dynamics at the late training phase. Instead, we expect that the regularization term can be controlled in a relatively small range for a long time while continuously increasing. As opposed to the linear increased regularization, we argue that using a nonlinear increased regularization term is more beneficial to the expressiveness of the winning tickets. As shown in Fig. 3.3, we propose several nonlinear regularization schemes: (1) **Logarithmic-Law (LogLaw):** $R = \frac{1}{ln2} \cdot R_{ceil} \cdot ln\left(\frac{\varphi}{E} + 1\right)$; (2) **Exponential-Law (ExpLaw):**

$R = \frac{1}{e-1} \cdot R_{ceil} \cdot \left(e^{\frac{\varphi}{E}} - 1\right)$; (3) **Tangent-Law (TanLaw):** $R = \frac{1}{2} \cdot R_{ceil} \cdot \left[\tan\left[\frac{\pi \cdot \varphi}{2E} - \frac{\pi}{4}\right] + 1\right]$. $R_{ceil}$ is the ceiling value of the regularization term. $\varphi$ is the current epoch and $E$ is the total epochs. Different growth strategies can increase the degrees of freedom in such regularization methods, thereby enabling us to obtain more diverse optimal subnetworks.

## 4 EXPERIMENTS

In this section, we evaluate our pruning algorithms (UniLTH and UniDLTH) on several widely-used datasets. Our goal is to answer the following research questions. **RQ1:** How effective do our algorithms find and transform winning lottery tickets (Sec 4.1)? **RQ2:** How does the subnetwork perform under different pruning rates (Sec 4.1)? **RQ3:** Is nonlinear increased regularization more powerful than its linear counterpart (Sec 4.2)? **RQ4:** How should we specify the detection the patience (people typically define a patience, i.e. the number of epochs to wait before early stop if no progress on the validation set.) $\phi$ of early stopping and the rewind epochs $\xi$ (Sec 4.3)? **RQ5:** Can our algorithms generalize to more backbones and larger-scale datasets (Appendix E)?

**Datasets and Backbones.** We first evaluate our algorithms on CIFAR10/100 datasets Krizhevsky et al. (2009) using VGG-19 Simonyan & Zisserman (2014) or ResNet-50 He et al. (2016) as the backbone. To further verify its effectiveness on large-scale datasets, we conduct experiments on ImageNet Simonyan & Zisserman (2014) using the backbone ResNet-50. Meanwhile, we consider the following methods for comparisons: 1) $L_1$ is the $L_1$ regularization pruning based on a pre-trained network Li et al. (2016). 2) **LTH/LTH-Iter** is the lottery ticket hypothesis based on one-shot/iterative pruning strategies Frankle & Carbin (2018). 3) **EB** is the Early-Bird ticket (i.e., LT which emerge at the very early training phase) for LTH with an one-shot pruning strategy Frankle & Carbin (2018). 4) **DLTH/DLTH-Iter** is the dual lottery ticket hypothesis based on one-shot/iterative pruning strategies Bai et al. (2022). Our experiments are run on two NVIDIA Tesla V100 GPUs. Due to the page limit, we describe the experimental settings in Appendix D.

### 4.1 PRUNING ALGORITHMS COMPARISON

In this part, we explore the efficacy of our proposed algorithms. For LTH, different iterative pruning values were proposed for comparison. For EB, we follow its original paper to set the early stopping point at 37 epochs (1/8 of the total epochs). For DLTH, we set weight decay starting from 0, and ceiling bound is 2e-3. Here we just adopt linear growth regularization for comparison and nonlinear regularization will be discussed in Sec. 4.2. In Tab. 1, we report the mean top-1 accuracy with its standard deviation of three-run experiments and we have placed notations in Appendix D.

**Performance Comparison (RQ1).** From Tab. 1, we have the following observations: (1) In most cases, our algorithms outperform other approaches on the four datasets in terms of both accuracy and standard deviation; (2) Using our training paradiam on the training process, the parameters of a large model (e.g., ResNet50, MobileNet) are heavily slashed (even by a factor of 10) while obtaining an excellent subnetwork under the same sparse condition (e.g., 93.09%/92.96% and 73.88%/73.68% of Vgg-19+CIFAR10 and ResNet-50+CIFAR100 settings); (3) Iterative pruning can help us get better lottery tickets. As seen in Tab. 1, UniLTH (Iter-10) mostly outperforms UniLTH (one-shot) among various backbones counterpart with datasets experimental settings. Similarly, we can draw the same conclusions in UniDLTH algorithm. These results experimentally demonstrate that iterative pruning allows our model to gain benefits to get better sub-networks. (4) Similar to standard LTH and DLTH, our algorithm can considerably improve the inference speed without significant performance drop.

**Results of Different Pruning Rate (RQ2).** Iterative pruning, early stopping with rewind, and gradually increased regularization are three key components of our algorithms. To answer RQ2, we

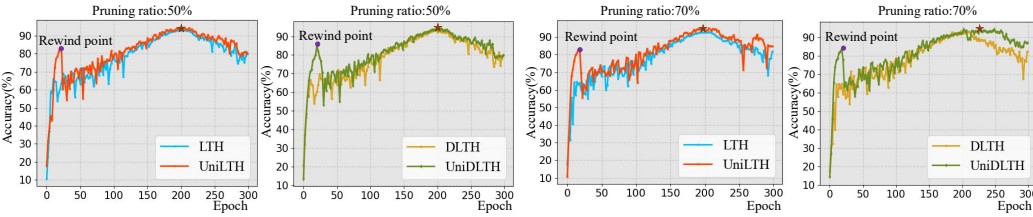

Figure 3: ResNet-50 on CIFAR-10 using 50%/70% sparsity with UniLTH/UniDLTH algorithms.

Table 1: Performance comparison of Vgg-19/ResNet50 on CIFAR10/CIFAR100 datasets using 30%, 50%, 70% and 90% sparsity ratios. Except for the $L_1$ row, the highest/second highest performances are emphasized with red/blue fonts

| Vgg19+CIFAR10; Baseline accuracy: 92.19% | | | | Vgg19+CIFAR100; Baseline accuracy: 56.33% | | | |
|---|---|---|---|---|---|---|---|
| Pruning ratio(%)/Speedup | 30%/1.43× | 50%/2.0× | 70%/3.33× | 90%/10.0× | 30%/1.43× | 50%/2.0× | 70%/3.33× | 90%/10.0× |
| $L_1$(pre-trained model) | 90.99±0.09 | 89.92±0.17 | 87.74±0.26 | 84.27±0.35 | 56.21±0.21 | 54.39±0.37 | 50.47±1.23 | 49.77±1.47 |
| LTH(one-shot) | 90.28±0.09 | 93.01±0.21 | 92.37±0.39 | 92.08±0.54 | 56.34±0.22 | 56.37±0.32 | 56.09±0.43 | 55.98±0.56 |
| LTH(Iter-10) | 93.29±0.16 | 93.15±0.38 | 92.77±0.23 | 92.34±0.25 | 56.51±0.17 | 56.53±0.23 | 56.32±0.27 | 56.38±0.28 |
| EB | 92.89±0.14 | 91.44±0.27 | 90.41±0.26 | 90.33±0.47 | 56.32±0.11 | 55.89±0.29 | 55.77±0.33 | 55.43±0.42 |
| DLTH(one-shot) | 92.94±0.18 | 92.87±0.44 | 92.16±0.51 | 92.07±0.49 | 56.23±0.19 | 56.21±0.21 | 55.87±0.38 | 55.71±0.53 |
| DLTH(Iter-10) | 92.95±0.12 | 92.78±0.22 | 92.84±0.31 | 92.40±0.37 | 56.37±0.13 | 56.43±0.14 | 56.29±0.25 | 56.04±0.29 |
| UniLTH(one-shot) | 93.34±0.06 | 93.21±0.38 | 93.08±0.25 | 92.88±0.28 | 56.47±0.09 | 56.42±0.27 | 55.98±0.23 | 56.01±0.36 |
| UniLTH(Iter-10) | 93.40±0.04 | 93.45±0.27 | 93.28±0.18 | 93.09±0.14 | 56.50±0.08 | 56.73±0.15 | 56.27±0.18 | 56.44±0.24 |
| UniDLTH(one-shot) | 93.27±0.13 | 93.20±0.26 | 92.89±0.34 | 92.72±0.35 | 56.48±0.14 | 56.43±0.31 | 56.21±0.35 | 55.78±0.49 |
| UniDLTH(Iter-10) | 93.52±0.11 | 93.48±0.17 | 93.19±0.28 | 92.96±0.21 | 56.77±0.13 | 56.68±0.24 | 56.47±0.27 | 56.41±0.29 |
| ResNet50+CIFAR10; Baseline accuracy: 93.45% | | | | ResNet50+CIFAR100; Baseline accuracy: 73.87% | | | |
| Pruning ratio(%)/Speedup | 30%/1.43× | 50%/2.0× | 70%/3.33× | 90%/10.0× | 30%/1.43× | 50%/2.0× | 70%/3.33× | 90%/10.0× |
| $L_1$(pre-trained model) | 93.28±0.17 | 92.17±0.24 | 90.34±0.31 | 87.25±0.42 | 72.98±0.16 | 70.26±0.35 | 69.30±0.47 | 68.43±0.42 |
| LTH(one-shot) | 92.68±0.17 | 92.78±0.36 | 92.38±0.29 | 92.29±0.43 | 72.38±0.14 | 72.66±0.27 | 72.45±0.33 | 72.37±0.36 |
| LTH(Iter-10) | 93.56±0.15 | 93.38±0.21 | 93.40±0.23 | 93.21±0.37 | 73.27±0.09 | 73.66±0.11 | 73.42±0.23 | 73.49±0.28 |
| EB | 92.89±0.21 | 91.56±0.38 | 89.73±0.32 | 89.69±0.35 | 73.13±0.18 | 72.88±0.24 | 72.04±0.35 | 72.16±0.63 |
| DLTH(one-shot) | 93.42±0.25 | 93.37±0.36 | 93.17±0.43 | 92.87±0.46 | 72.88±0.17 | 72.67±0.29 | 72.84±0.51 | 72.71±0.47 |
| DLTH(Iter-10) | 93.57±0.17 | 93.69±0.23 | 93.50±0.15 | 93.29±0.28 | 73.67±0.04 | 73.23±0.14 | 73.41±0.25 | 73.22±0.30 |
| UniLTH(one-shot) | 93.63±0.33 | 93.58±0.22 | 93.26±0.38 | 93.18±0.36 | 73.77±0.18 | 73.60±0.33 | 73.19±0.43 | 73.41±0.52 |
| UniLTH(Iter-10) | 93.71±0.27 | 93.66±0.15 | 93.58±0.19 | 93.69±0.22 | 73.91±0.07 | 73.89±0.20 | 73.75±0.22 | 73.68±0.18 |
| UniDLTH(one-shot) | 92.87±0.38 | 93.46±0.29 | 93.52±0.34 | 93.28±0.40 | 73.10±0.19 | 73.12±0.21 | 72.89±0.38 | 73.11±0.37 |
| UniDLTH(Iter-10) | 94.15±0.21 | 93.58±0.20 | 93.62±0.17 | 93.48±0.17 | 74.05±0.12 | 73.98±0.14 | 73.60±0.12 | 73.88±0.26 |

control the number of pruning iterations $\psi$ to 10, the monitoring threshold $\phi$ (i.e., patience) for the descent of validation loss to be 2, and the rewind epoch to be 2 to keep these variables consistent. As shown in Fig. 3, we validate our methods under 50%/70% pruning ratio on combination settings of ResNet-50 and CIFAR-10. It can be seen that our unified algorithms (i.e., UniLTH and UniDLTH) clearly surpass the traditional algorithms on both pruning rates, which demonstrates their effectiveness. Additionally, we observe in Fig. 3 that at the early stage of training, the abandonment of regularization can help the network to learn data distribution faster.

## 4.2 LINEAR VERSUS NONLINEAR INCREASED REGULARIZATION

In this paper, we propose three forms of nonlinear increased regularization (including LogLaw, TanLaw and ExpLaw) with different regularization intensities at the early, middle and late stage, respectively. To answer **RQ3**, we compare them with the linear counterpart and meanwhile investigate the effect of different regularization force ceiling $R_{ceil}$ on the model performance.

In Tab. 2, when using $R_{ceil}$ with the same magnitude, ExpLaw significantly outperforms the other three strategies under all pruning ratios. As mentioned before, increased regularization stimulates the diversity of parameters. Different growth strategies (linear or nonlinear) further increase the degrees of freedom in such regularization methods, i.e., enabling us to obtain more diverse optimal subnetworks. For example, LogLaw keeps the network under intense regularization during the whole training process (see Fig.2), which

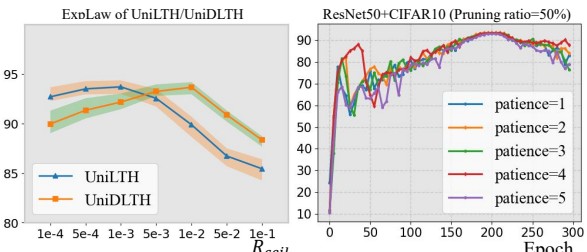

Figure 4: (*Left*) ExpLaw on ResNet-50 + CIFAR-10 + 50% sparsity experimental settings under different $R_{ceil}$. (*Right*) Network performance under different validation loss monitor threshold $\phi$ (i.e., patience).

might lead to ill-conditioned weights. Conversely, ExpLaw utilizes smaller penalty for a long training time and can simultaneously ensure that the weights magnitude discrepancy will be magnified.

Table 2: Performance comparison of different increased regularization strategies on Vgg-19/CIFAR10 and ResNet-50/CIFAR100 experimental settings. For fairness, we set all $R_{ceil}$ to be consistent and keep at 1e-3 (UniLTH/LTH) and 1e-2 (UniDLTH/DLTH).

| Settings | | | Pruning ratio(%)/Speedup | | | |
|---|---|---|---|---|---|---|
| Datasets/Backbones | Algorithms | Regularization | 30%/1.43× | 50%/2.0× | 70%/3.33× | 90%/10.0× |
| CIFAR 10/ Vgg-19 | UniLTH (Iter-10) | Linear | 93.40±0.16 | 93.45±0.27 | 93.28±0.18 | 93.09±0.14 |
| | | LogLaw | 91.89±0.12 | 91.77±0.17 | 90.79±0.31 | 91.04±0.43 |
| | | TanLaw | 92.44±0.18 | 92.35±0.23 | 91.48±0.25 | 91.21±0.39 |
| | | ExpLaw | **93.98±0.24** | **93.79±0.26** | **93.48±0.27** | **93.22±0.31** |
| | UniDLTH (Iter-10) | Linear | 93.52±0.11 | 93.48±0.17 | 93.19±0.28 | 92.96±0.21 |
| | | LogLaw | 91.57±0.18 | 91.32±0.48 | 91.03±0.44 | 89.99±0.31 |
| | | TanLaw | 92.48±0.32 | 92.36±0.39 | 91.49±0.35 | 91.15±0.51 |
| | | ExpLaw | **93.80±0.27** | **93.65±0.34** | **93.77±0.29** | **93.29±0.43** |
| CIFAR 100/ ResNet-50 | UniLTH (Iter-10) | Linear | 73.91±0.07 | 73.89±0.20 | 73.75±0.22 | 73.68±0.18 |
| | | LogLaw | 71.17±0.34 | 71.21±0.36 | 70.66±0.47 | 70.83±0.41 |
| | | TanLaw | 72.17±0.20 | 72.05±0.34 | 72.15±0.44 | 71.62±0.53 |
| | | ExpLaw | **74.14±0.36** | **74.29±0.32** | **74.77±0.28** | **74.47±0.30** |
| | UniDLTH (Iter-10) | Linear | 74.05±0.12 | 73.98±0.14 | 73.60±0.12 | 73.88±0.26 |
| | | LogLaw | 71.12±0.15 | 71.29±0.22 | 70.37±0.35 | 70.27±0.46 |
| | | TanLaw | 72.21±0.17 | 72.24±0.38 | 71.79±0.41 | 71.62±0.53 |
| | | ExpLaw | **74.18±0.15** | **74.09±0.24** | **74.64±0.35** | **74.38±0.37** |

Moreover, the ill-conditioned weight issue can also be prevented by means of a small penalty, which is more conducive to the search/transformation of excellent subnetworks. We further explore the impacts of different $R_{ceil}$ on the model results. As depicted in the left hand side of Fig. 4, we find that UniLTH requires less penalty force than UniDLTH. This may be attributed to the uncertainty of the subnetwork structure at the beginning of LTH and the need to slowly drop weights for a reliable subnetwork. Excessive regularization force will destroy the learning process and complicate the search of winning tickets. In contrast, the discarded parameters in UniLTH are given at the beginning, which requires a larger gradually increasing regularization force to extrude the information as much as possible, so $R_{ceil}$ large is allowed.

### 4.3 EFFECTS OF PATIENCE FOR EARLY STOPPING AND REWIND VALUES

To answer **RQ4**, we first set patience $\phi$ to 2 and monitor the top-1 accuracy under various rewind epochs $\xi$. After that, we fix $\xi$ as 2 and test network performance under different early stopping patience $\xi$ under 30%, 50%, 70% and 90% pruning ratio (denoted by $p$). Related results are reported in Tab. 3 and Fig. 4 (Right), from which we have the following findings. The model exhibits strong expressiveness when $\xi$ is controlled in a small regime such as 1 or 2. In contrast, when enlarging the rewind epochs to some extent, e.g., $\xi = 4, 5$,

Table 3: Top-1 accuracy of UniLTH vs. rewind epochs $\xi$ and early-stopping patience $p$. We adopt CIFAR10+Vgg-19 for evaluation and set $\phi = 2$.

| Pruning ratio | $\xi = 1$ | $\xi = 2$ | $\xi = 3$ | $\xi = 4$ | $\xi = 5$ |
|---|---|---|---|---|---|
| $p = 30\%$ | 93.44 | 93.65 | 92.98 | 91.34 | 89.45 |
| $p = 50\%$ | 93.48 | 93.87 | 92.68 | 92.01 | 90.55 |
| $p = 70\%$ | 93.45 | 93.64 | 93.35 | 92.17 | 90.11 |
| $p = 90\%$ | 93.65 | 93.66 | 93.44 | 92.45 | 90.24 |

the model performance will be degenerated by a considerable margin. Based on such observations, smaller $\xi$ are recommended to avoid crippling the model capacity. Meanwhile, we find that different $\phi$ does not have much effect on final model performance. Although the early train performance of the model was different in various patience, the model always reached the optimum at nearly 200 epochs with similar accuracy (Seen Fig. 4 (Right)).

## 5 CONCLUSION

In this work, we present a unified paradigm for searching and transforming winning lottery tickets. At the early training phase, we replace regularization with early stopping to mitigate overfitting and ensure the stability of training. When parameter distribution reaches a certain point, we rewind weights and alternately adopt increased regularization and pruning to search/transform subnetworks to achieve network capacity and sparsity joint-optimization. In addition, to compensate for the ill-conditioned small-weight problem caused by linearly increased regularization, a variety of nonlinear regularization methods are proposed, among which ExpLaw method improves the network expression power by limiting the network role to a small penalty force for a long time. We have benchmarked our algorithms on extensive datasets and backbones, achieving comparable performance on ultra-lightweight subnetworks.

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

## A    PROOF OF EQ. 8

In Eq. 7, the early stopping is equivalent to $L_2$ regularization when $(\Lambda + \alpha I)^{-1}\alpha = (I - \epsilon\Lambda)^{\tau}$. We take the logarithm on both sides as:

$$\log[(\Lambda + \alpha I)^{-1}\alpha] = \log(I - \epsilon\Lambda)^{\tau} \quad \rightarrow \quad \log(diag[\frac{\alpha}{\lambda_1 + \alpha}, ..., \frac{\alpha}{\lambda_1 + \alpha}]) = \tau \cdot \log(I - \epsilon\Lambda) \quad (9)$$

Substituting $\frac{\alpha}{\lambda_i + \alpha} = (1 + \frac{\lambda_i}{\alpha})^{-1}$ into Eq. 9, we can obtain:

$$-\log\left(I + \frac{\Lambda}{\alpha}\right) = \tau\log(I - \epsilon\Lambda) \tag{10}$$

According to the Taylor's Expansion, when $x$ approaches zero, $\log(1-x)$ can be approximated as $x$. Likewise, when the eigenvalues in $\Lambda$ approach zeros, it can be seen that $-\log(I + \Lambda/\alpha) \approx \Lambda/\alpha$ and $\tau\log(I - \epsilon\Lambda) \approx \tau\epsilon\Lambda$. Therefore, the equivalence holds if $\alpha \approx \frac{1}{\epsilon\tau}$.

## B    PROOF OF $L_2$ REGULARIZATION IS EQUIVALENT TO EARLY STOPPING WHEN $w^{(0)} \neq 0$

When $w^{(0)} \neq 0$, we have:

$$Q^T w^{(\tau)} = (I - \epsilon\Lambda)^{\tau} Q^T w^{(0)} + [I - (I - \epsilon\Lambda)^{\tau}] Q^T w^* \tag{11}$$

This equation can also be written as:

$$Q^T w^{(\tau)} = \left[(I - \epsilon\Lambda)^{\tau} \frac{w^{(0)}}{w^*} + I - (I - \epsilon\Lambda)^{\tau}\right] Q^T w^* = \left[I - (I - \epsilon\Lambda)^{\tau}\left(I - \frac{w^{(0)}}{w^*}\right)\right] Q^T w^* \tag{12}$$

In practice $w^{(0)}$ is usually much smaller than $w^*$, which means that $\frac{w^{(0)}}{w^*}$ approaches 0 (i.e., $I - \frac{w^{(0)}}{w^*} \approx I$). Compare Eq. 12 with Eq. 7, the equivalence still holds if $\alpha \approx \frac{1}{\epsilon\tau}$.

## C   PROOF OF INCREASED REGULARIZATION

**Increased Regularization.** As the formula $\acute{w} = (H + \alpha I)^{-1}Hw^*$ shown, after the regularization force increases (increase $\delta\alpha$ on the original basis), the parameters in the network will exercise to a new position. When the network converges again, it still exists $\acute{w} = (H + \delta\alpha I)^{-1}Hw^*$. Due to $(H + \delta\alpha I)^{-1}$ is very complicated or even difficult to solve, to better analyze the changes of the parameters in the network after the regularization force increases, we simply explore two H forms of H forms (we investigate two simplify cases to help us move forward) Wang et al. (2020).

(1) $H$ is diagonal matrix, which is a simplest form for Hessian information LeCun et al. (1989). We assume $H = \text{diag}(h_{1,1}, \ldots, \text{diag}\, h_{n,n})$, then:

$$\acute{w} = (H + \delta\alpha I)^{-1}Hw^* = \text{diag}\left(\frac{h_{1,1}}{h_{1,1} + \delta\alpha}, \cdots, \frac{h_{n.n}}{h_{n.n} + \delta\alpha}\right)w^* \tag{13}$$

We can find that for $i$-th $w_i^*$ in $w^*$, $\frac{\acute{w}_i}{w_i^*} = \frac{h_{i,i}}{h_{i,i}+\delta\alpha} \in [0,1)$ since $h_{i,i} > 0$ and $\delta\alpha > 0$. It is not difficult to find a larger curvature corresponding to the larger hessian element. The closer the ratio of $\frac{\acute{w}_i}{w_i^*}$ is to 1, the less the weight moves.

(2) $H$ is not diagonal. Here we analysis 2d matrix, $w^* = \begin{pmatrix} w_1^* \\ w_2^* \end{pmatrix}$ and $H = \begin{pmatrix} h_{1,1} & h_{1,2} \\ h_{2,1} & h_{2,2} \end{pmatrix}$, then:

$$\begin{pmatrix} \acute{w}_1 \\ \acute{w}_2 \end{pmatrix} = \frac{1}{|H + \delta\alpha I|}\left\{\begin{matrix}(h_{1,1}h_{2,2} + h_{1,1}\delta\alpha - h_{1,2}^2)w_1^* + \delta\alpha h_{1,2}w_2^* \\ (h_{1,1}h_{2,2} + h_{2,2}\delta\alpha - h_{1,2}^2)w_2^* + \delta\alpha h_{1,2}w_1^*\end{matrix}\right\} \tag{14}$$

Due to $\delta\alpha$ is small, we can obtain:

$$\begin{pmatrix} \acute{w}_1 \\ \acute{w}_2 \end{pmatrix} \approx \frac{1}{|H + \delta\alpha I|}\left\{\begin{matrix}(h_{1,1}h_{2,2} + h_{1,1}\delta\alpha - h_{1,2}^2)w_1^* \\ (h_{1,1}h_{2,2} + h_{2,2}\delta\alpha - h_{1,2}^2)w_2^*\end{matrix}\right\} \tag{15}$$

It is not difficult to find that at $h_{1,1} > h_{2,2}$, we can still get $\frac{\acute{w}_1}{w_1^*} > \acute{w}_2/w_2^*$ similar conclusion. To summarize, due to different local second-order partial derivative structures, different weight responses are different in response to increased regularization forces. Larger curvature results in the weights being relatively less moved towards the original points. The magnitude of the difference among the weights will increase as regularization grow.

## D   EXPERIMENTAL SETTINGS AND NOTATIONS

The experiments are conducted on two NVIDIA Tesla V100 (16GB per GPU) and all selected backbones follow the same experimental settings for fairness. Specifically, experiments on CIFAR10/CIFAR100 are optimized by Stochastic Gradient Descent (SGD), and we control the learning rate to 0.1 with 0.9 momentum using batch size 128. We use the Cosine Annealing Warm Restarts Loshchilov & Hutter (2016) scheduler as our learning rate scheduler and set the maximum number of iterations $T_{max} = 200$. Those models for ImageNet classification are optimized by SGD and lr=0.1 with 0.9 momentum and we follow the same setting in learning rate scheduler.

Table 4: The notations commonly reported in this work are placed here.

| Notation | Definition |
|---|---|
| $X$ | Input data |
| $\Theta$ | Network parameters |
| $m_\Theta$ | Mask matrix |
| $p\%$ | Pruning rate |
| $\psi$ | Pruning times |
| $\phi$ | Validation patience |
| $\xi$ | Rewind epochs |
| $R_{ceil}$ | The ceiling of regularization term |

# E    EXPERIMENTS ON MORE BACKBONES AND LARGE-SCALE DATASETS

To answer **RQ5**, we here evaluate MobileNets (v1) Howard et al. (2017) and EfficientNetB0Tan & Le (2019) two backbones on CIFAR-10 datasets, and add additional ImageNet dataset to verify the performance of our proposed unified lottery ticket search/transform algorithm. The results are given in the table below.

Table 5: Performance comparasion of different pruning algorithm counterpart with MobileNets(v1) and EfficientNetB0 backbones on CIFAR 10 dataset using 30%, 50%, 70% and 90% sparsity ratios. For convenience, we highlight show the highest/second highest performances with red/blue fonts.

| MobileNets+CIFAR 10; Baseline accuracy: 91.11% | | | | |
|---|---|---|---|---|
| pruning ratio(%)/speedup | 30/1.43× | 50/2.0× | 70/3.33× | 90/10.0× |
| L1(pre-trained model) | 90.21±0.09 | 89.37±0.25 | 88.77±0.41 | 87.21±0.52 |
| LTH(one-shot) | 90.23±0.38 | 90.16±0.47 | 89.91±0.52 | 88.73±0.60 |
| LTH(Iter-10) | 91.37±0.25 | 90.67±0.29 | 91.11±0.32 | 89.16±0.37 |
| EB | 91.47±0.23 | 90.16±0.34 | 89.96±0.41 | 89.25±0.39 |
| DLTH(one-shot) | 90.98±0.34 | 90.16±0.33 | 89.65±0.48 | 89.31±0.58 |
| DLTH(Iter-10) | 91.27±0.18 | 91.10±0.26 | 89.49±0.34 | 89.99±0.40 |
| UniLTH(one-shot) | 91.17±0.24 | 90.89±0.17 | 90.04±0.25 | 89.90±0.33 |
| UniLTH(Iter-10) | 92.14±0.07 | 91.18±0.15 | 91.12±0.19 | 90.24±0.26 |
| UniDLTH(one-shot) | 91.24±0.21 | 90.44±0.12 | 90.01±0.32 | 89.52±0.27 |
| UniDLTH(Iter-10) | 92.07±0.11 | 91.20±0.14 | 90.10±0.21 | 89.97±0.25 |
| EfficientNetB0+CIFAR 10; Baseline accuracy: 90.64% | | | | |
| pruning ratio(%)/speedup | 30/1.43× | 50/2.0× | 70/3.33× | 90/10.0× |
| L1(pre-trained model) | 90.72±0.19 | 89.88±0.21 | 89.02±0.33 | 87.16±0.59 |
| LTH(one-shot) | 90.29±0.32 | 90.16±0.41 | 89.87±0.50 | 89.72±0.68 |
| LTH(Iter-10) | 90.66±0.25 | 90.21±0.19 | 89.98±0.37 | 89.96±0.34 |
| EB | 90.27±0.37 | 89.98±0.46 | 89.45±0.62 | 88.93±0.73 |
| DLTH(one-shot) | 90.44±0.31 | 90.27±0.45 | 90.10±0.39 | 89.65±0.47 |
| DLTH(Iter-10) | 90.79±0.22 | 90.66±0.38 | 90.09±0.25 | 90.11±0.35 |
| UniLTH(one-shot) | 90.37±0.27 | 90.23±0.33 | 90.16±0.42 | 89.26±0.37 |
| UniLTH(Iter-10) | 90.77±0.19 | 90.79±0.25 | 90.23±0.26 | 90.34±0.22 |
| UniDLTH(one-shot) | 90.44±0.34 | 90.21±0.41 | 89.40±0.39 | 89.18±0.58 |
| UniDLTH(Iter-10) | 91.26±0.37 | 90.14±0.24 | 90.18±0.25 | 90.42±0.31 |

The results of comparative evaluation experiments on MobileNets(v1) and EfficientNetB0 backbones are summarized in Table 5. Our proposed pruning algorithm achieves almost the best performance under the same pruning ratios, which demonstrates the effectiveness of our unified winning lottery tickets search/transform algorithms in a more general scenario.

Table 6: Performance on ImageNet dataset counterpart with different backbones and **UniLTH** pruning algorithm using 30%, 50%, 70% and 90% sparsity ratios. Acc@1: If the highest probability is the correct answer, it is considered correct. Acc@5: A correct answer is considered correct if the top five probabilities contain the correct answer.

| Different backbones + UniLTH: ImageNet | | | | | | | | |
|---|---|---|---|---|---|---|---|---|
| Pruning ratio(%)/speedup | 30/1.43× | | 50/2.0× | | 70/3.33× | | 90/10.0× | |
| Metrics | Acc@1 | Acc@5 | Acc@1 | Acc@5 | Acc@1 | Acc@5 | Acc@1 | Acc@5 |
| Vgg19 (baseline Acc@1=71.98, Acc@5=90.18) | 72.66 | 90.47 | 72.35 | 90.35 | 72.17 | 90.24 | 71.63 | 89.96 |
| ResNet50 (baseline Acc@1=74.72, Acc@5=91.87) | 74.98 | 92.03 | 74.86 | 92.19 | 73.96 | 92.07 | 73.88 | 91.44 |
| MobileNets (baseline Acc@1=70.77,Acc@5=88.43) | 71.34 | 90.17 | 70.86 | 89.52 | 71.05 | 89.11 | 70.69 | 88.26 |
| EfficientNet (baseline Acc@1=76.28,Acc@5=92.96) | 76.96 | 93.14 | 76.77 | 93.17 | 76.23 | 92.45 | 75.89 | 92.34 |

From Table 6 and Table 7, we can observe that when using the larger-scale dataset ImageNet, our unified lottery pruning algorithm is still effective, and when the model is pruned to one tenth of the original network, the backbones performance can still be comparable to the original backbones excellent performance. To summarize, We conclude our UniLTH/UniDLTH are also adaptable and robust for general backbones and large-scale datasets and even obtains better performances using the same parameter setting compared with LTH/DLTH, whcih can further search/transform the ultra-lightweight subnetworks.

Table 7: Performance on ImageNet dataset counterpart with different backbones and **UniDLTH** pruning algorithm using 30%, 50%, 70% and 90% sparsity ratios. Acc@1: If the highest probability is the correct answer, it is considered correct. Acc@5: A correct answer is considered correct if the top five probabilities contain the correct answer.

| Different backbones + UniDLTH: ImageNet | | | | | | | | |
|---|---|---|---|---|---|---|---|---|
| Pruning ratio(%)/speedup | 30/1.43× | | 50/2.0× | | 70/3.33× | | 90/10.0× | |
| Metrics | Acc@1 | Acc@5 | Acc@1 | Acc@5 | Acc@1 | Acc@5 | Acc@1 | Acc@5 |
| Vgg19 (baseline Acc@1=71.98, Acc@5=90.18) | 72.25 | 90.28 | 71.98 | 90.66 | 72.13 | 90.33 | 71.41 | 89.87 |
| ResNet50 (baseline Acc@1=74.72, Acc@5=91.87) | 74.96 | 92.14 | 74.75 | 92.23 | 73.84 | 92.11 | 73.81 | 91.5 |
| MobileNets (baseline Acc@1=70.77,Acc@5=88.43) | 71.41 | 89.13 | 70.72 | 89.24 | 71.47 | 89.23 | 70.48 | 88.22 |
| EfficientNet (baseline Acc@1=76.28,Acc@5=92.96) | 76.97 | 93.07 | 76.76 | 92.98 | 76.45 | 92.08 | 76.07 | 92.41 |

