# OpenReview forum: "A UNIFIED VIEW OF FINDING AND TRANSFORMING WINNING LOTTERY TICKETS"
_ICLR.cc/2023/Conference — Submitted to ICLR 2023_

### Official Review · Reviewer_vpf2 · 2022-10-23

**Confidence:** 4
**Clarity, Quality, Novelty And Reproducibility:** The paper is well-written. The idea i…
**Correctness:** 4
**Technical Novelty And Significance:** 3
**Empirical Novelty And Significance:** 3
**Recommendation:** 6

**Strength And Weaknesses:**

Strengths:

1. This paper conducts experiments on diverse datasets and architectures. The experiments are thorough and convincing.
2. The paper provides clear motivation and complete theoretical proof of the early stopping strategy.

Weaknesses:

1. As mentioned in Section 1, regularization-based pruning approaches (e.g., DLTH) typically perform one-shot pruning, which exacerbates the instability of sparse network training. Nevertheless, it seems no results to show how unstable the DLTH method is.

2. This paper mainly compares the accuracy on several benchmarks and shows that the resultant winning tickets often outperform the full network. It would be interesting to see whether the winning ticket is also more robust than the full model against common corruptions (e.g., on CIFAR-10-C or CIFAR-100-C).

3. There are some missing related work [A,B] that also focus on finding and exploiting winning/losing tickets. It would be better to discuss the differences between the proposed approach and these methods from the perspective of how to find subnetworks of interest.


4. This paper mainly compares the proposed method with LTH and DLTH. It is interesting to see how large the improvement is compared with some popular one-shot pruning methods, such as ThiNet [C] and DCP [D].

5. Apart from the improved pruning results, training cost is another important factor to evaluate a method. Will the proposed method introduce additional training cost compared with LTH and DLTH? If so, how much cost is introduced?

[A] A winning hand: Compressing deep networks can improve out-of-distribution robustness, NeurIPS 2021.
[B] Improving robustness by enhancing weak subnets, ECCV 2022.
[C] Thinet: A filter level pruning method for deep neural network compression, ICCV 2017.
[D] Discrimination-aware channel pruning for deep neural networks, NeurIPS 2018.


**Summary Of The Paper:**

The proposed UniLTH and UniDLTH pruning methods are well-motivated. The authors provide a new understanding of the regularization in the early stage. The paper is well-written and provides rigorous mathematical proof, which is a good contribution to the community. In the main paper and supplementary materials, extensive experiments with different sparsity levels on diverse tasks also verify the effectiveness of the proposed method.

**Summary Of The Review:**

The proposed method is well-motivated. The authors provide a new understanding of regularization along with rigorous proof. Thus, I tend to vote accept.

---

### Official Review · Reviewer_FGiK · 2022-10-24

**Confidence:** 4
**Correctness:** 3
**Technical Novelty And Significance:** 2
**Empirical Novelty And Significance:** 2
**Recommendation:** 3

**Clarity, Quality, Novelty And Reproducibility:**

As discussed in the previous section, I have some serious concerns regarding the novelty and the clarity of this paper.

**Strength And Weaknesses:**

I have several concerns regarding this paper. First of all, I think that the technical contribution of the proposed method is limited: the increased regularization term was already presented in the DLTH, instead a pretraining phase is already present in the procedure of the lottery ticket hypothesis. I have also some doubts regarding the regularization term: it is well known that the l1 norm is more effective than the l2 norm in terms of sparsification, therefore it is not clear to me why the authors choose to use the l2 norm as regularization term. Another concern is related to the English usage: the paper is not well written, there are many grammar and syntax errors making the paper hard to follow.

**Summary Of The Paper:**

This paper extends the dual lottery ticket hypothesis (DLTH), which consists in randomly selecting a subnetwork and transforming it in a winning ticket through training with regularization. In this paper, the authors propose s slightly modified procedure, called UniDLTH, where they introduce a preliminary training phase without regularization. The authors also propose a novel procedure for the lottery tickets hypothesis where they introduce a gradually increased regularization term.

**Summary Of The Review:**

I think that this paper presents some serious issues. In particular, the novelty and the presentation quality are below the level expected for ICLR.

---

### Official Review · Reviewer_3BTa · 2022-10-27

**Confidence:** 4
**Correctness:** 2
**Technical Novelty And Significance:** 2
**Empirical Novelty And Significance:** 2
**Recommendation:** 3

**Clarity, Quality, Novelty And Reproducibility:**

**Clarity, Quality and Novelty**
- As stated under weaknesses, the writing needs to be improved significantly.
- The idea is somewhat novel, but I am not convinced of the importance of combining these two algorithms into a single framework. Especially since the performance improvements are marginal.

**Strength And Weaknesses:**

**Strenghts:**
- Their method combines two complementary views of finding lottery tickets: (i) IMP, the traditional mechanism of pruning a neural network to find subnetworks that can be trained to full accuracy, (ii) Dual LTH (Bai et al.), which "transforms" a random subnetwork so that it can be trained to full accuracy through regularization.
- They conduct experiments on multiple datasets including CIFAR-10, CIFAR-100 and Imagenet.

**Weaknesses:**
1. I'm not sure it is important to combine the LTH and Dual LTH into a single framework. That said, UniLTH and UniDLTH don't seem to perform particularly well against their counterparts and the comparisons seem to be incomplete.
    - Consider Table 1: Is baseline dense training? If so, how is it worse than sparse training? If this is true, then the hyperparameters of the different algorithms have not been tuned properly. Also, the accuracy seems to typically increases with sparsity, which again does not make sense.
    - Since the DLTH is not a pruning at init method, you should be comparing against pruning after training schemes as well. In which case, you should compare against Renda et al. (Comparing Rewinding and Fine-tuning in Neural Network Pruning) which performs significantly better than IMP even with warmup.
    - Table 2: LogLaw and TanLaw regularization is always worse than Linear and ExpLaw is only marginally better. This makes me wonder if hyperparameters were tuned for all of them. Moreover, it does not really show that nonlinear regularization is important which is the claim of the work.
    - Figure 3: I think it would be beneficial to include other algorithms in this plot. This plot merely tells me that LTH and UniLTH are almost the same. Also, Figure 3 is not mentioned in the text, so I am not sure what inference I am supposed to make from it.
2. The writing needs to be improved. Several paragraphs are convoluted and hard to read.
    - Please use \citep{} when including multiple citations which are not part of the sentence. For example, in the 3rd line of the introduction: "... though over-parameterized deep neural networks achieve encouraging performance over widespread machine learning tasks Zagoruyko & Komodakis (2016); Arora et al. (2019); Devlin et al. (2018); Brown et al. (2020), they usually suffer notoriously ...". The citations should be in parenthesis otherwise it is difficult to read.
    - Page 2: analogy -> analogous
    - LTH is used synonymously with IMP as an algorithm in this paper. This is incorrect. LTH is a claim regarding the existence of sparse subnetworks. Iterative Magnitude Pruning (IMP) is the algorithm you are referring to which is commonly used to find Lottery Tickets. There are several other algorithms: Refer SNIP (Tanaka et al.), GraSP (Wang et al.), Continuous Sparsification (Savarese et al.) to mention a few.
    - In Section 2, I believe Malach et al. is cited incorrectly. Their result does not include any mention of training. It is purely an approximation result known as the Strong Lottery Ticket Hypothesis.
    - Section 2 is also missing several references: Ramanujan et al., Renda et al., Su et al., Sreenivasan et al.
    - Section 3.2 is confusing. You claim to prove that early stopping is equivalent to regularization but then say that early stopping is better. Isn't this a contradiction? I aalso do not agree with the claim that early stopping is "non-parametric". You still need to choose some thresholds on what constitutes convergence of validation loss as well as choice of the validation set.
    - Section 4.1: paradiam -> paradigm.
3. Questions about the algorithm:
    - In Section 3, it is stated that training is performed until validation loss stops dropping. And then the parameters are rewound to $\xi$ epochs earlier. I don't follow this step. Why train if you are going to rewind all of the weights? Are these trained weights used in some way before rewinding? This is not clear in the algorithm.
    - What is $\psi$ in Section 2? It is not defined.
    -  In Algorithm 1, line 5: what does "several epochs ago" mean? Please be precise.
    -  Is the nesting in Algorithm 1 correct? If *if* condition in line 15 is False, then the *while* loop in line 1 will also be false. Both these things do not need to exist. Please clarify/rewrite Algorithm 1. I'm not sure I follow it.
4. Result about early stopping is equivalent to regularization:
    - You assume that $J(w)$ reaches its minimum claiming that this is the condition for early stopping. However, this is the true objective, not validation loss. If you assume that these two are equivalent, then early stopping is essentially stopping at the minimizer of the true function, which is incorrect.
    - Analyzing gradient descent on $\hat{J}$ to approximately study gradient descent on $J$ needs to be justified more. Firstly, the approximation only holds in a small neighborhood and it is not clear that GD will stay in that neighborhood. Moreover, this is a quadratic function and it is well known that GD concerges exponentially fast for quadratics. This is surely not true for the function you are considering which might be highly non-convex.
    - Typo: satisfies -> satisfied.
    - Please rephrase the paragraph on "Why Early Stopping is preferable": It is quite convoluted.
    - Appendix B: What does $\frac{w^{0}}{w^*}$ mean? If they are both vectors, then I am not sure what dividing them even means. Are you conisdering norms? Furthermore, I don't agree that $w^{0}$ is usually much smaller than $w^*$. If you are training with regularization, the expectation is that the norm of $w$ will decrease as training proceeds.




**Summary Of The Paper:**

The paper presents a unified method to combine both iterative magnitude pruning and the Dual Lottery Ticket Hypothesis (Bai et al.) by combining early stopping and regularization in addition to traditional magnitude pruning. They also prove that early stopping is equivalent to $L_2$ regularization and also suggest some novel nonlinear regularization schemes. They conduct experiments on CIFAR-10 and Imagenet to back their claims.

**Summary Of The Review:**

I have several concerns about the writing and the algorithm itself. If those are clarified, then I would reconsider but in its current form, I do not recommend acceptance. I am also not convinced of the value of combining these two algorithms into a single framework.

---

### Official Review · Reviewer_4vCa · 2022-11-02

**Confidence:** 2
**Correctness:** 3
**Technical Novelty And Significance:** 3
**Empirical Novelty And Significance:** 3
**Recommendation:** 6

**Clarity, Quality, Novelty And Reproducibility:**

Overall, the paper is easy to follow and well-presented. The exploration of different nonlinear regularization methods is novel to me; however, the effectiveness of using early stopping and "the unified view" remains unclear.

**Strength And Weaknesses:**

Pros:
- It is interesting and well-demonstrated to find a connection between the $\ell_2$ weight regularization and early stopping by providing an approximate condition between the hyperparameters governing weight decay and SGD update.
- The study on different nonlinear increasing regularization functions seems like a novel direction to the community. It will be interesting to learn nonlinear functions to dynamically control the whole regularization growing process –– which may avoid the early stopping yet unify the two-stage in the proposed method as an adaptive nonlinear regularization function.
- The experiment was well-designed and conducted. Extensive experimental results and analyses were provided to validate the proposed methods in several aspects.

Cons:
- It is not clear to me how the proposed method is called a unified approach/view over LTH and DLTH. I may miss something; yet I did not find a unified framework/formulation for these two complementary methods. It seems like the proposed methods simply adopt early stopping in these two methods, respectively. By using a linear regularization function, is UniLTH (UniDLTH) w/o early stopping the same to the LTH (UniLTH)?
- While the connection between weight decay and early stopping is interesting, it is not well-motivated why using early stopping can benefit the training process, especially when they could be equivalent by tuning the learning rate and pre-training steps (given by Eq (8)). This is also demonstrated by the limited improvement of testing accuracy on CIFAR 10 and CIFAR100.
- Some necessary experimental results are missing -- 1) the ablation of the proposed method on early stopping and nonlinear regularization functions; 2) the comparison results with other baselines on the ImagNet dataset; 3) the proposed method under 5 iterative pruning and a more aggressive pruning ratio.

**Summary Of The Paper:**

The paper revisits the LTH and DLTH by relaxing the weight regularization in the early training stage, which tries to adopt early stopping instead to realize a better pre-trained weight initialization. Two sparse network training methods, termed as UniLTH and UniDLTH, were designed and developed through a nonlinear increasing weight regularization method. Extensive experimental results were provided on two public datasets with several backbones.

**Summary Of The Review:**

Please refer to my comments in "Strength And Weaknesses" for more details.

---

> ### Author Response · Authors · 2022-11-13
> **Response to Reviewer 4vCa:**
>
> We sincerely appreciate reviewer 4vCa for the thoughtful feedback. We are excited that review 4vCa affirms the novelty, clarity, and easy to follow of our work. Below, we reply to the concerns and questions raised by reviewer 4vCa:
>
> > **Comment 1. It is not clear to me how the proposed method is called a unified approach/view over LTH and DLTH. I may miss something; yet I did not find a unified framework/formulation for these two complementary methods. It seems like the proposed methods simply adopt early stopping in these two methods, respectively. By using a linear regularization function, is UniLTH (UniDLTH) w/o early stopping the same to the LTH (UniLTH)?**
>
> Thank you for pointing out this issue, and sorry for that we didn’t explain this well. Lottery Ticket Hypothesis (LTH) tries to find an admirable subnetwork in a full network—like winning tickets in lottery pool. Dual Lottery Ticket Hypothesis (DLTH) considers a complementary case—*transforming a randomly selected subnetwork into an admirable one*. The key idea of these two works are iterative pruning and linear increased regularization.
>
> In our work, we combine these two tools into our UniLTH/UniDLTH framework and take an in-depth view of the relationship between $L_{2}$ regularization and early stopping, and the role of the non-linear increased regularization. Finally, we formalize two-stage learning paradigms to identify (UniLTH) or transform (UniDLTH) a winning ticket. The “unified” can been seen in following three perspectives (seen Figure 1. (b) and Algo 1.):
>
> + **Unified early stopping policy**: In the first stage, we do not introduce any regularization force at the early training phase to ensure that the model accurately learns the training data distribution. Instead, we perform training, early stopping, and rewinding at the first stage.
> + **Unified pruning strategy**: we perform iterative pruning strategy on both UniLTH and UniDLTH
> + **Unified increased regularization paradigm**: we argue that using a nonlinear increased regularization term is more beneficial to the expressiveness of the winning tickets. And we replace linear increased regularization with non-linear increased regularization in the second stage.
>
> The difference is only in the way we define the subnet. UniLTH does not select the “remain” substructure of the full model before training and attaches non-linear increased regularization on all weights, while UniDLTH pre-defines the substructure and applies gradually increasing regularization to the discarded part to extrude information to the complementary part.
>
>
> > **Comment 2. While the connection between weight decay and early stopping is interesting, it is not well-motivated why using early stopping can benefit the training process, especially when they could be equivalent by tuning the learning rate and pre-training steps (given by Eq (8)). This is also demonstrated by the limited improvement of testing accuracy on CIFAR 10 and CIFAR100.**
>
> Sorry for this confusion, [1][2] have argued that early stopping has the effect of restricting the optimization procedure to a relatively small volume of parameter space in the neighborhood of the initial parameter value. As shown in Figure 1. (b) and according to the proof of our work, a trajectory of length $\varpi$ can ends at a point that corresponds to a minimum of the $L_{2}$ regularized objective. We summarize benefit of early stopping as:
>
> Different from $L_{2}$ regularization, early stopping typical involves monitoring the **validation set (non-parameterized)** error in order to stop trajectory at a particularly good point in space in the first stage.
>
> In practical usage, early stopping **silently saves the training computational** consume in that it automatically determines the correct amount of regularization while weight decay requires many training experiments with diﬀerent values of its hyperparameter (tune the hyperparameters) [3].
>
> [1] Bishop C M. Regularization and complexity control in feed-forward networks[J]. 1995.
> [2] Sjöberg J, Ljung L. Overtraining, regularization and searching for a minimum, with application to neural networks[J]. International Journal of Control, 1995, 62(6): 1391-1407.
> [3] LeCun Y, Bengio Y, Hinton G. Deep learning[J]. nature, 2015, 521(7553): 436-444.
>
> > **Comment 3. Some necessary experimental results are missing.**
>
> We agree! We have tested the effect of on early stopping and nonlinear regularization functions in Tab 2 and Tab 3. Meanwhile, we have added the comparison results with other baselines (Vgg-19, ResNet- 50, MobileNets, and EfficientNet) on the ImagNet dataset.

---

### Decision · Program_Chairs · 2023-01-20

**Decision:**

Reject

**Justification For Why Not Higher Score:**

Authors only respond to one of the four reviewers and do not answer any questions raised by the reviewers who were negative.

**Justification For Why Not Lower Score:**

N/A

**Metareview: Summary, Strengths And Weaknesses:**

This paper presents a pruning and regularization approach to training sparse neural networks, cast within the lottery ticket hypothesis framework.

Reviewer opinion is split, with two marginal accept and two reject ratings among four reviewers.  Unfortunately, the authors only responded to Reviewer 4vCa (who gave a marginal accept), and leave questions and concerns raised by Reviewers 3BTa, FGiK, and vpf2 unanswered.  In recommending reject, Reviewers 3BTa and FGiK were concerned over clarity of presentation, and the value and technical contribution of the proposed algorithmic framework.  Reviewer vpf2 asks for clarification on training costs, and points out missing related work which should be cited and compared.  These are all significant concerns, and without any attempt by the authors to address them, the Area Chair cannot recommend acceptance.